# LEARNING TO ACT THROUGH ACTIVATION FUNCTION OPTIMIZATION IN RANDOM NETWORKS

## ABSTRACT

Biological neural networks are characterised by a high degree of neural diversity, a trait that artificial neural networks (ANNs) generally lack. Additionally, learning in ANNs is typically synonymous with only modifying the strengths of connection weights. However, there is much evidence from neuroscience that different classes of neurons each have crucial roles in the information processing done by the network. In nature, each neuron is a dynamical system that is a powerful information processor in its own right. In this paper we ask the question, how well can ANNs learn to perform reinforcement learning tasks only through the optimization of neural activation functions, without any weight optimization? We demonstrate the viability of the method and show that the neural parameters are expressive enough to allow learning three different continuous control tasks without weight optimization. These results open up for more possibilities for synergies between synaptic and neural optimization in ANNs in the future. Code is available from [anonymised].

## 1 INTRODUCTION

Artificial neural networks (ANNs) have been shown to be able to learn a wide variety of different tasks (Schmidhuber, 2015). With inspiration from their biological counterparts (Hassabis et al., 2017), ANNs have dramatically pushed the boundaries for what is achievable for artificial intelligence technologies. ANNs are trained by tuning a large number of parameters, each of which provides a small contribution to the final output of the network. Likewise, most, but not all (Titley et al., 2017), learning and behavioral changes are manifested in the biological brain as long- or short-term potentiation or depression of synapses between neurons (Stiles, 2000).

Neurons of the human brain are characterised by a high degree of diversity (Lillien, 1997; Soltesz et al., 2006), and different classes of neurons respond differently to the incoming signals (Izhikevich, 2003). A single biological neuron is a sophisticated processor in its own rights (Izhikevich, 2007; Poirazi et al., 2003), with information processing occurring at several steps both in its dendrites (Beaulieu-Laroche et al., 2018; Magee, 2000), cell body and axon terminals (Kamiya & Debanne, 2020; Rama et al., 2018). Neurons of various classes interconnect in intricate circuits (Breedlove & Watson, 2013; Kandel et al., 2000). This suggests that at least part of the explanation behind the impressive ability of biological networks to learn and retain knowledge must be found in the interplay between the abundance of different neuron types (Nusser, 2018).

While the diversity of biological neurons is well documented, in ANNs it is common to have a single activation function used by all hidden neurons. Intrigued by the interesting properties of randomly-initialised networks in both machine learning (Gaier & Ha, 2019; Najarro & Risi, 2020; Ulyanov et al., 2018) and neuroscience (Lindsay et al., 2017), we are interested in the computational expressivity of only optimizing parameterized neural activation functions without any weight optimization. As described below, our approach allows every neuron in our ANNs to be a unique dynamical system.

We apply our method to three diverse continuous control tasks. The simpler `CartPoleSwingUp` Gaier & Ha (2019), the locomotion of a bipedal robot (Brockman et al., 2016), and a vision-based car racing task with procedurally generated tracks (Brockman et al., 2016). The results show that the method performs well on all three tasks, outperforming weight-optimized networks that have a

similar number of adjustable parameters in two out of three tasks. Surprisingly, optimized activation functions in random networks even outperform a weight-optimized network with many more adjustable parameters in the more challenging `CarRacing-v0` environment.

While there exist previous work that deals with optimizing activation function parameters (Agostinelli et al., 2014; Sipper, 2021; Bingham & Miikkulainen, 2022), to the best of our knowledge, here we show for the first time that optimizing expressive activation functions alone allows solving challenging RL tasks. We hope that our results inspire further research in approaches that do not only see ANN optmization as the optimization of weight parameters and challenging some of our assumptions on what it means for such systems to learn.

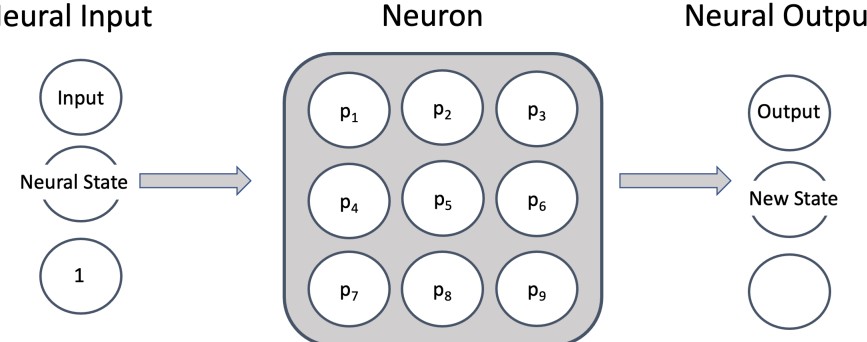

Figure 1: Illustration of the proposed neural activation function. The parameters $p_i$ are optimized in order to achieve an expressive function. These parameters are used to integrate the input with a neural state and a bias term through a vector-matrix multiplication. As input, the neuron takes an input value propagated from the previous layer, its neural state, and a constant value of one as a bias term. It outputs a value to be propagated to the next layer, as well as its own new state. For further details see Section 3.

## 2 RELATED WORK

**Neurocentric Optimization.** Biases in neural networks is an example of neurocentric parameters. When an ANN is optimized to solve any task, the values of the network's weights and biases are gradually being tuned until a functional network has been achieved. The network most commonly has a one bias value for each neuron and the weight parameters thus greatly outnumber these neurocentric bias parameters. It is well known that the function of biases is to translate the activations in the network (Benítez et al., 1997), and ease the optimization of the network. Another well known example of neurocentric parameters are found in the PReLU activation functions (He et al., 2015) where a parameter is learned to determine the slope of the function in the case of negative inputs. Introducing this per neuron customization of the activation functions was shown to improve performance of networks with little extra computational cost. Neurocentric parameter optimization can also be found within the field of plastic neural networks. In one setting of their experiments, Urzelai & Floreano (2001) optimized plasticity rules for each neuron (they referred to this as 'node encoding'), such that each incoming synapse to a node was adapted by a common plasticity rule. The idea of neurocentric parameters is therefore far from new. However, in contrast to earlier work, in this paper we explore the potential of solely optimizing the activation functions of a randomly initialized network without ever adapting the weights.

**Activation Functions in Neuroevolution.** Not all ANNs have a single activation function for all hidden neurons. Some version of Neuro-Evolution of Augmented Topologies (NEAT) (Stanley & Miikkulainen, 2002; Papavasileiou et al., 2021) allow for different activation functions on each neuron. The NEAT algorithm searches through networks with increasing complexity over the process of evolution. Starting from a simple network structure, each new network has a chance of adding a new neuron to the network. When a new neuron is added, it can be allocated a random activation function from a number of predetermined functions. In newer versions of NEAT, mutations allow activation functions of neurons to be changed even after it was initially added (Hagg et al., 2017). This resulted

in more parsimonious networks. In their *weight agnostic neural network* (WANN) work, Gaier & Ha (2019) used NEAT to find network structures that could perform well, even when all weights had the same value. Notably, as part of the evolved structure was the possibility of neurons to have different activation functions, and different neurons could thus respond differently to the same incoming signals. This likely extended the expressiveness of the WANNs considerably. We see the work presented below as being in similar spirit as that of Gaier and Ha, in that we are also exploring the capabilities of a component of neural networks in the absence of traditional weight optimization. In the work presented in this paper, all networks have a standard fully connected structure. Further, we do not choose from a set of standard activation functions, but introduce stateful neurons with several parameters to tune for each individual activation function.

**Lottery Tickets & Supermasks.** The Lottery Ticket Hypothesis (Frankle & Carbin, 2018) states that the reason large ANNs are trainable is that a large network contains a combinatorial number of subnetworks, one of them is likely to be easily trainable for the task at hand. The hypothesis is based on the finding that after having trained a large network, it is usually possible to prune a large portion of the parameters without suffering significant loss in performance (Li et al., 2016). An even stronger take on the Lottery Ticket Hypothesis states that due to the sheer number of subnetworks that are present within a large network, it is possible to learn a binary mask on top of the weight matrices of a randomly initialized neural network, and in this manner get a network that can solve the task at hand (Malach et al., 2020; Wortsman et al., 2020; Ramanujan et al., 2020). This has even been shown to be possible at the level of neurons; with a large enough network initialization, a network can be optimized simply by masking out a portion of the neurons in the network (Wortsman et al., 2020; Malach et al., 2020). Learning an activation function for each neuron could be seen as learning a sophisticated mask on top each neuron, something that has been demonstrated to be possible. However, the idea of learning a mask on top of a random network relies on the random network being large enough (Malach et al., 2020), so that the chance of it containing a useful subnetwork is high. Masking neurons is a less expressive mask method than masking weights: it is equivalent to masking full columns of the weight matrices instead of strategically singling out weights in the weight matrix. As such, the method of masking neurons require larger random networks in order to be successful. In this paper, we optimize neurons' activation functions in relatively small networks. This is possible because the activation functions themselves are much more expressive than a binary mask.

## 3 SEARCHING ONLY FOR ACTIVATION FUNCTIONS IN RANDOM NETWORKS

Typically, optimization of ANNs has been framed as the learning of distributed representations (Bengio et al., 2013) that can become progressively more abstract with the depth of the network. Optimization of weights, is a process of fine-tuning the iterative transformation of one representation into another in order to end with a new, more useful representation of the input. How the intermediate layers respond to a given input depends on the specific configuration of the weight matrix responsible for transforming the input as well as their activation function.

Randomly-initialised networks can already perform useful computations (Ulyanov et al., 2018; He et al., 2016; Hochreiter & Schmidhuber, 1997). When activation functions are trained specifically to interpret signals from a fixed random matrix, as is the case in this paper, the initially arbitrary transformations will become meaningful to the function, as long as there exist detectable patterns between the input the function receives and the output the function passes on and is evaluated on. Whether a pattern is detectable depends on the expressiveness of the function. With this in mind, it is reasonable to assume that if activation functions of the neurons in the network are made more expressive, they can result in useful representations even when provided with arbitrary transformations of the input.

Motivated by the diversity of neuron types in animal brains (Soltesz et al., 2006), we aim to test how well RL agents can perform if we only optimize the parameters of a network's activation functions while not optimizing its weights at all. An illustration of the activation function representation we optimize in this paper is shown in Fig. 1. Each activation function consist of a small three-by-three matrix of values to be optimized. Representing a neuron by a small matrix means that the neuron can take more than a single value as input, as well as outputting more than one value. We utilize this to endow each neuron with a state. The state of the neuron is integrated with the input

through the optimized neural parameters. Part of the neuron's output becomes the new state of the neuron, which is fed back to the neuron with the next input. This turns our activation functions into dynamical systems. Presented with the same input value at different points in the neuron's history can thus yield different outputs. We find that stateful neurons provide a convenient and efficient way of equipping a network with some memory capabilities.

When a representation has been transformed by a weight matrix, each activation function receives one value of the transformation, same as in traditional ANNs. This input value is concatenated with the current state of the neuron and a third value that is always equal to one. Together, these form a vector. The output of a neuron is the vector-matrix multiplication of this vector and the matrix that makes up the parameterized neuron. From the perspective of a single neuron, this can be written as:

$$[\hat{x}_{l,i}^t, h_{l,i}^t, o] = tanh(\mathbf{n}_{l,i} \cdot [x_{l,i}^t, h_{l,i}^{t-1}, 1]^T). \tag{1}$$

Here, $x_{l,i}^t$ is the input value, $h_{l,i}^{t-1}$ is the state of the neuron, and $\mathbf{n_{l,i}}$ the neuron with $l$ denoting the current layer in the network, $i$ the placement of the neuron in the layer, and $t$ is the current time step. A value of 1 is concatenated as a bias term for the neuron. The hyperbolic tangent function is used for non-linearity and to restrict output values to be in [-1, 1]. $\hat{x}_{l+1,j}^t$ is the value that is propagated through weights connecting to the subsequent layer, and $h_{l,i}^t$ is the updated state of the neuron. $o$, the third value in the output of the neuron is discarded. As the three-by-three matrix has nine values in total, we need to optimize nine parameters for each neuron in the network.

## 4 EXPERIMENTS

We optimize activation function parameters in otherwise standard fully connected feedforward neural networks. All the these networks have two hidden layers, containing 128 and 64 neurons, respectively. We use learned activation functions on all neurons, including in the input and output layer. The sizes of the input and output layers vary with the environments described below. The fixed weight values are sampled from $\mathcal{N}(0, 0.5)$. We ran each experiment three times on different seeds, excepts for the weight-optimized models in the environment, which we ran twice due to its longer training times.

As baselines, we optimize weights of standard feedforward networks. For these we run two different settings: one has a similar number of adjustable parameters as the number of parameters in the activation function approach (**Small FFNN Architecture**). In order to get the number of weights to be similar to activation function parameters, the widths and depths of these networks have to be considerably smaller than the random networks used in the main experiments. In the second baseline setting (**Same FFNN Architecture**), we train weights of networks that have the same widths and depths as the networks in the main experiments, and thus many more adjustable parameters. Unless stated otherwise, the activation function for all baseline experiments is the hyperbolic tangent function for all neurons.

### 4.1 ENVIRONMENTS

We test the effectiveness of only optimizing the parameters of the activation functions in randomly initialized networks in three diverse continuous control tasks: the CartPoleSwingUp environment (Gaier & Ha, 2019), the Bipedal Walker environment, and the Car Racing environment (Brockman et al., 2016), which are described below:

**CartPoleSwingUp.** This environment is a variation of the classic control task (Barto et al., 1983), where a cart is rewarded for balancing a pole for as long as possible. In the `CartPoleSwingUp` variation, an episode starts with the pole hanging downwards, and in order to score points, the agent must move the cart such that the pole gets to an upright position. From there, the task is to keep balancing the pole. We use the implementation of Gaier and Ha (Gaier & Ha, 2019). The agent gets five values as input, and must output a single value in [-1,1] in order to move the cart left and right. With these input- and output layers, the total number of neurons in the network becomes 198, and we optimize 1,792 parameters. For a feedforward network with a similar number of weights we optimize a network with two hidden layers of size 48 and 32, respectively.

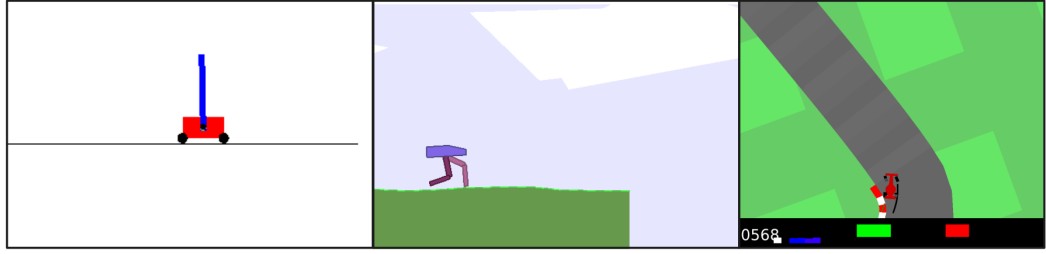

Figure 2: Environments used for experiments. Left: `CartPoleSwingUp`. Middle: `BipedalWalker-v3`. Right: `CarRacing-v0`.

Table 1: Convolutional Layer Parameters.

|  | Layer 1 | Layer 2 |
| --- | --- | --- |
| Input Channels | 3 | 6 |
| Output Channels | 6 | 8 |
| Kernel Size | 3 | 5 |
| Stride | 1 | 2 |
| Activation Function | tanh | tanh |
| Bias | Not used | Not used |

**Bipedal Walker.** To get a maximum score in the `BipedalWalker-v3` environment (Brockman et al., 2016), a two-legged robot needs to learn to walk as efficiently and robustly as possible. The terrain is procedurally generated with small bumps that can cause the robot to trip if its gait is too brittle. Falling over results in a large penalty to the overall score of the episode. Observations in this environment consist of 24 values, including LIDAR detectors and information about the robots joint positions and speed. For actions, four values continuous values in $[-1, 1]$ are needed. This result in a network with 220 neurons altogether. To get a network with a similar number of weights as there are parameters in the main experiment, we train a feedforward network with two hidden layers, both containing 32 neurons.

**Car Racing.** In the `CarRacing-v0` domain (Brockman et al., 2016), a car must learn to navigate through procedurally generated tracks. The car is rewarded for getting as far as possible while keeping inside the track at all times. To control the car, actions consisting of three continuous values are needed. One of these is in $[-1, 1]$ (for steering left and right), while the other two are in $[0, 1]$, one for controlling the gas, the other for controlling the break. The input is a top down image of the car and its surroundings, as shown in Fig. 4.1. For the experiments in this paper, we wanted to focus on the effectiveness of the activation functions in fully connected networks. We therefore followed a strategy closely mimicking that used by Najarro & Risi (2020) to get a flat representation of the input image. We normalize the input values and resize the image into the shape of $3x84x84$. The image is then send through two convolutional layers, with the parameters of these specified in Table 1. After both layers a two-dimensional max pooling with kernel size 2 and stride 2 was used to gradually reduce the number of pixels.

The output of the convolutional layers is flattened, resulting in a vector containing $648$ values. This vector is then used as input to a fully connected feedforward network with our proposed activation functions. Importantly, the parameters of the convolutional layers stay fixed after initialization and are never optimized. The output layer has three neurons. With the much larger input layer, this network has 844 neurons and 7,596 adjustable parameters. Since two of the action values should be in $[0, 1]$, a sigmoid function is used for these two output neurons in the place of the hyperbolic tangent function as shown in Equation 1.

For the baseline experiments, we use the same strategy of convolutional layers with fixed parameters to get a flat input for the feedforward networks, the weights of which we are optimizing. Since the input is so large, the feedforward network can only have a single hidden layer of size 12 to get a similar number of adjustable parameters as in the main approach. This results in a network with 7,827 parameters, including weights and biases. The activation function for all neurons in the

network is the hyperbolic tangent function, except for two of the output neurons, which are activated by the sigmoid function.

## 4.2 OPTIMIZATION DETAILS

For parameter optimization, we use a combination of a Genetic Algorithm (GA) and Covariance Matrix Adpatation Evolution Strategy (CMA-ES) Hansen (2006). More specifically, we GA is used for the first 100 generations of the optimization. The best solution found by the GA is then used as starting point for the CMA-ES algorithm. For both algorithms, we use off-the-shelf implementations provided by Ha (2017a) and Hansen (2006). For all experiments, the GA uses a population size of 512, and its mutations are drawn from a normal distribution $\mathcal{N}(0, 1)$. All other hyperparameters are the default parameters of the implementation. The large sigma means that the GA can cover a large area, but in a coarse manner. The use of an initial GA phase is meant to kick-start the optimization for subsequent fine-tuning by CMA-ES. For the CMA-ES algorithm, we use a population size of 128, and set weight decay to zero. Other than that, hyperparameters are the default parameters of the implementation. The total number of generations is 1,400 for the `Car Racing` environment, and 4,000 for the `BipedalWalker` and `CartPoleSwingUp` environments.

For the large weight-optimized networks, CMA-ES becomes impractical to use due to its use of the covariance matrix of size $N^2$ with $N$ being the number of parameters to optimize. For this reason, we instead use Evolution Strategy (sometimes referred to as "OpenES" (Ha, 2017b)) as described in Salimans et al. (2017) and. We use a population size of 128 and otherwise use the default parameters of the implementation (Ha, 2017b).

## 5 RESULTS

Evaluations of the most successful runs of each experimental setting are summarized in Table 2. All experimental settings achieved good scores on the `CartPoleSwingUp` task. Only the weight-optimized network with hidden layers of size 128 and 64 managed to average a score above 300 points over 100 episodes in the `BipedalWalker-v3` environment. However, the activation functions alone came close with just a fraction of the number of optimized parameters. The smaller weight-optimized network's average score of 235 is indicative of the agent having learned to walk to the end of the level in most cases, but in an inefficient manner.

In the `CarRacing-v3` environment, our method scored the highest, though none of the approaches reached an average score above 900 over 100 episode, which is needed for the task to be considered solved. Training curves for all experimental settings can be found in Appendix A.1.

### 5.1 INVESTIGATING LEARNED ACTIVATION FUNCTIONS

To get a better idea how a neural network with random weights but optimized activation functions is solving the task, we plotted all activation functions of a layer of the champion network (Fig. 5.2) of the `CarRacing` environment. The figure shows that while several of the found activation functions look like the standard form we would expect from a hyperbolic tangent function with a bias, many of the other types of functions also emerged after optimization. We see functions with strong oscillatory behaviors, some in the whole input space, and some only in smaller sections. Other functions have extra peaks and valleys compared to the standard hyperbolic tangent. This allows these functions to respond with more nuance to input and create richer representations. Analysing these networks further to get a more detailed understanding of how the solve the task is an interesting future research direction.

### 5.2 COMPARISON TO WEIGHT AGNOSTIC NETWORKS

An approach similar in spirit to ours is the weight agnostic neural network approach by Gaier & Ha (2019). As detailed in Section 2, in WANNs, only the architecture of the neural network is learned (including choosing an activation function from a predefined set for each neuron) while avoiding weight training. While an apples-to-apples comparison is not possible (due to different optimization algorithms), it is interesting to see how these two methods compare. The optimized activation functions tend to score better in all three environments. Since parameters are added to the

Table 2: Table of Results. Means and standard deviations over 100 episodes, and number of parameters for each experimental setting. In the 'Activation Functions' models, only parameters of activation functions are optimized. In the other models, only the weight parameters of feedforward neural networks are optimized. Scores are evaluated with the most successful run in each setting. For context, results from Weight Agnostic Neural Networks (WANN) (Gaier & Ha, 2019) are included as another method that does not optimize weights.

|  | Model | Score | # Params. |
|---|---|---|---|
| CartPoleSwingUp | Activation Functions | $916 \pm 79$ | **1,782** |
|  | Small FFNN Architecture | **927** $\pm 83$ | 1,889 |
|  | Same FFNN Architecture | $922 \pm 73$ | 9,089 |
|  | WANN | $732 \pm 16$ | N/A |
| BipedalWalker | Activation Functions | $295 \pm 63$ | **1,980** |
|  | Small FFNN Architecture | $235 \pm 13$ | 1,988 |
|  | Same FFNN Architecture | **318** $\pm 46$ | 11,716 |
|  | WANN | $261 \pm 58$ | N/A |
| CarRacing | Activation Functions | **812** $\pm 95$ | **7,587** |
|  | Small FFNN Architecture | $770 \pm 167$ | 7,827 |
|  | Same FFNN Architecture | $752 \pm 171$ | 91,523 |
|  | WANN | $608 \pm 161$ | N/A |

WANN models during optimization, we cannot compare the numbers of adjustable parameters in these models to the that of the activation functions. However, these results suggest that it might be easier to optimize customizable activation functions for each position in a fully connected network, than it is to learn a network structure from scratch.

In the future both approaches could be complementary. We imagine that extending the WANN approach with more expressive activation functions could allow their evolved neural architectures to become significantly more compact and higher-performing.

## 6 DISCUSSION AND FUTURE WORK

In this paper, we introduced an approach to optimize activation functions for stateful neurons. Training these alone yielded neural networks that can control agents in the `CartPoleSwingUp`, `CarRacing`, and `BipedalWalker` environments, even when the weights of the network were never optimized. The largest weight-optimized network achieved superior scores compared to our method in the `CartPoleSwingUp` and `BipedalWalker` environments. This is not surprising; weight optimization of ANNs now has a long history of success in a plethora of domains. When using random transformations, there is a risk of getting a degraded signal, something that can be compensated for easily by tuning the weights of the transformations. Nevertheless, the proposed activation function alone did enable meaningful behavioral changes in the agents. Surprisingly, the optimized activation functions achieved the best score of the experimental settings in the `CarRacing` task. The failure of the larger weight-optimized network to perform well here might be explained by the relatively low population size compared to the number of parameters being optimized (128, and 91,523, respectively). This population size was the same for all experimental settings to ensure that all models are evaluated the same number of times in the environments during optimization. The advantage of having a smaller number of adjustable parameters also came into display in that the larger models could not be optimized by the more power CMA-ES method.

Our results come with some caveats. Specifically, they should be viewed as a proofs-of-concept lacking extensive hyperparameter search. Hyperparameters that are likely important for achieving better performance include network depth and layer width, as well as the distribution from which weights of the random networks are drawn. Here, we were mainly interested in showing performance of the activation functions in relatively small networks. Further, we only tested a few hand-selected distributions for the random weights in preliminary experiments before settling on the distributions used in the experiments described above. Therefore, these are areas were there is likely room improvements. As part of the proposed activation function, we included a persistent neural state that is fed back to the input of the neuron at the subsequent time step. This provides a endow the network

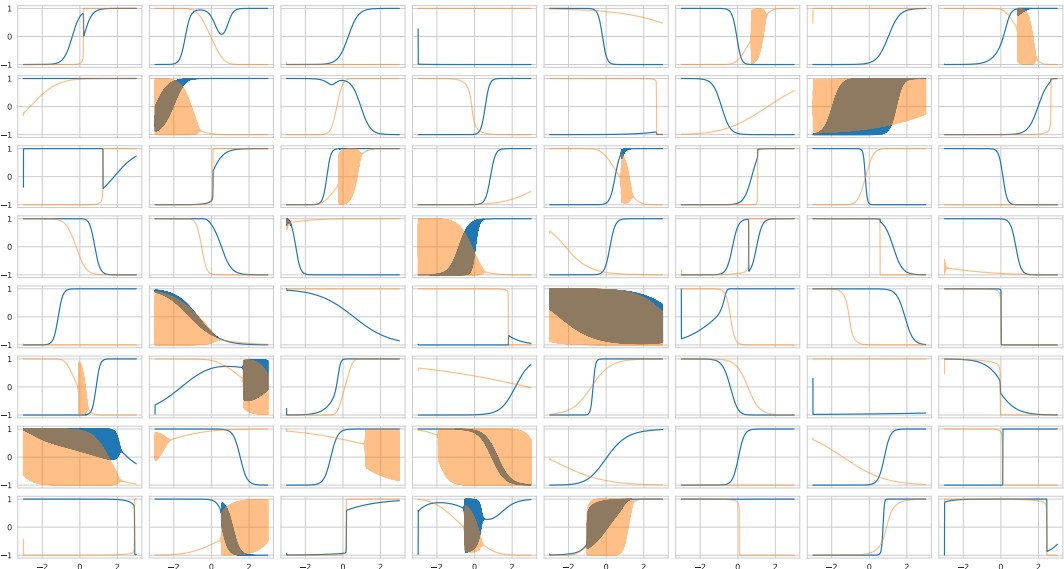

Figure 3: Diverse Activation Functions. Displayed are each of the 64 activation functions of the second hidden layer optimized to solve the `CarRacing` task. In blue are the activations that are passed on to the next layer. In orange are the neural states. One thousand inputs are given from -3 to 3 in an ordered manner, from most negative to most positive. Given the updatable neural state, the ordering of the inputs matter, and a different ordering would have yielded different plots.For all plots, the neural state is initialized as zero before the first input. Several of the found activation functions look like we would expect the hyperbolic tangent function with a bias and/or the possibility of a negative sign to look. Few functions seem unresponsive to the input. However, many functions are clearly both responsive and different from the standard form of the hyperbolic tangent function. We find functions that have oscillatory behavior in some or all of the input space. Other functions are non-monotonic and have peaks and valleys in particular areas.

with a memory mechanism. Memory as local neural states is unusual in ANNs, but is much more common in more the biologically inspired Spiking Neural Networks (SNNs) (Tavanaei et al., 2019; Pfeiffer & Pfeil, 2018; Izhikevich, 2006). Such a neural state is most useful for data with a temporal element such as agents acting in an environment. It is reasonable to assume that the same approach would have limited use in tasks with unordered data. However, for RL tasks, stateful neurons provide a relatively cheap way of allowing the network to have some memory capacity. Setting up traditional recurrent neural networks (RNNs), like LSTMs (Hochreiter & Schmidhuber, 1997), for the tasks used in this paper, would likely result the need of many more adjustable parameters than the number of parameters in our activation functions. Combining stateful neurons with RNNs could result in interesting memory dynamics on different timescales.

The ability to improve performance while leaving weights random opens up the possibility for future work of combining activation functions like the ones proposed here with the approach of masking weights Frankle & Carbin (2018) mentioned in Section 2. An advantage of using masks is that one can train masks for different tasks on the same random network (Wortsman et al., 2020). With pre-trained activation functions, it should be possible to bias the random network to perform actions that are generally useful within a specific task distribution. One could then train masks on top of the untouched weights in order to perform well on specific tasks, conceivably more efficient than with generic activation functions.

Another potentially interesting avenue for future work is to combine optimization of activation functions with synaptic plasticity functions Soltoggio et al. (2018). A lot of work has been done in the area of learning useful Hebbian-like learning rules (Chalmers, 1991; Miconi, 2016; Mouret & Tonelli, 2014; Floreano & Mattiussi, 2008; Najarro & Risi, 2020; Risi & Stanley, 2012; Soltoggio et al., 2008; Tonelli & Mouret, 2013; Wang et al., 2019). Less work in has explored the interaction

between learning rules and neural activation in ANNs, despite the fact that most learning rules take the neural activations as inputs. It seems likely that more expressive activation functions would in turn result in more expressive updates of weights via activity-dependent learning, and thus more powerful plastic neural networks.

Having shown that the proposed activation functions can achieve well-performing networks when optimized alone, a future experiments will explore the co-evolution of neural and synaptic parameters. It will be interesting to see whether a synergistic effect arises between these two sets of parameters. If both weights and activation functions are optimized together, will the there be as much diversity in the resulting set of activation functions, or will the need for such diversity decrease?

Despite being initially inspired by biological neural networks, ANNs are still far from their biological counter parts in countless aspects. While the work presented here does not claim to have presented a biologically plausible approach, we do believe that inspiration from biological intelligence still offers great opportunities to explore new variations of ANNs that can ultimately lead to interesting and useful results.

## ACKNOWLEDGEMENTS

Removed for blind review.

## REPRODUCIBILITY STATEMENT

The code to to reproduce every experiment in this paper is available at [removed].

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

# A APPENDIX

## A.1 TRAINING CURVES

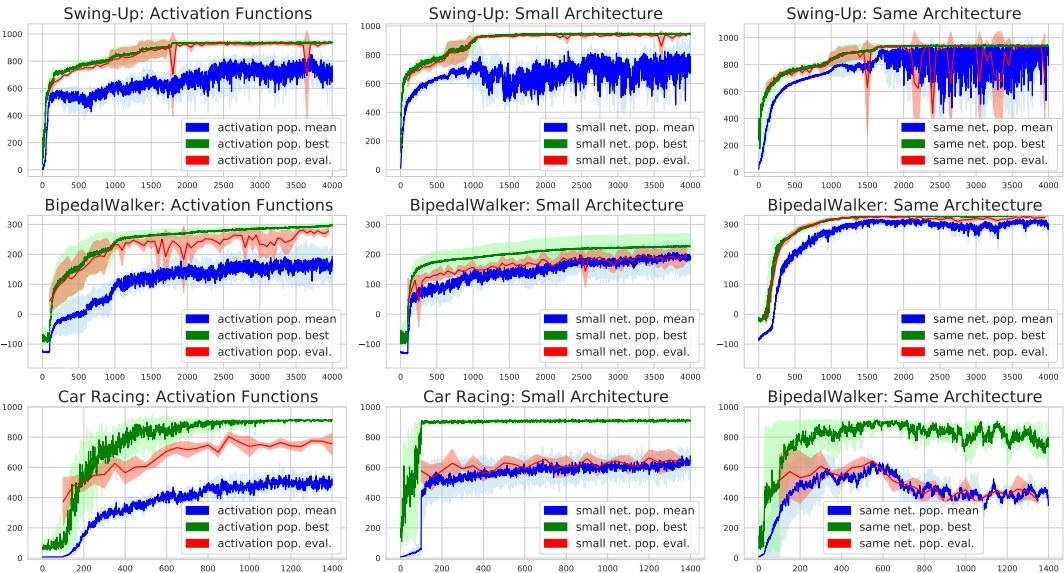

Figure 4: Training curves for all experimental settings. Means and standard deviations for populations of runs on different seeds. Each setting was run three times except for the weight-optimized models in the Car Racing envinronment, which were run twice. Every $50^{th}$ generation, the current solution was evaluated 64 times (orange).

