# OpenReview forum: "Learning to Act through Activation Function Optimization in Random Networks"
_ICLR.cc/2023/Conference — Submitted to ICLR 2023_

### Official Review · Reviewer_5XEX · 2022-10-16

**Confidence:** 3
**Correctness:** 3
**Technical Novelty And Significance:** 2
**Empirical Novelty And Significance:** 2
**Recommendation:** 5

**Clarity, Quality, Novelty And Reproducibility:**

The writing is generally clear. There are a few typos, e.g.:

- p. 5, "Car Racing": "3x84x84" has the wrong kind of 'x'.
- p. 6, Section 4.2: "More specifically, we GA..."
- Section 5: Should "CarRacing-v3" be "v0" ?
- Section 6: fix last sentence of first paragraph.


**Strength And Weaknesses:**

Strengths:

- The paper proposes an idea which I think is novel, and the result was not obvious a priori.

Weaknesses:

1) My main concern is that the rhetoric does not match the actual content of the paper. The paper does not do what it says on the tin! We are told that networks are to be trained "only through the optimization of neural activation functions". But the  proposed method  does much more than just "optimizing activation functions": it adds a recurrent neural state to the neurons! This is quite different from what would typically be understood  as "optimizing the activation function" alone, and  may be the main source of performance.

The impact of adding recurrence is particularly acute because all the experiments involve dynamic control  tasks, in which  introducing recurrence to otherwise purely feedforward network could  certainly provide an advantage. It is unlikely that the method would provide similar performance on tasks that don't have a time dimension, such as image classification!

2) Another concern is that of fair comparison. All networks  were trained with evolutionary algorithms (CMA-ES for the small ones, and plain ES for the large weight-optimized one). The authors claim that being able to use CMA-ES (due to the small number of trained parameters in comparison to the size of the architecture) is an advantage of the method.

But in reality, dynamic control tasks are more likely to be solved  with offline RL algorithms (like PPO or  SAC) rather than evolution.

I note that, because the proposed method introduces recurrence, it would be difficult to train with offline RL methods based on replay buffers and would  likely require policy-gradient methods like A3C, which are more stable but less sample-efficient. Thus, while the method does allow certain algorithms to be used, it also prevents the use of other algorithms, which are much more popular in the literature and more sample efficient by design.

How do the various architectures perform when trained with their best possible  algorithms? What would be the outcome of training  the weight-optimized networks with, say, SAC (or some other modern RL algorithm) rather than evolution?  Is it possible that the apparent competitive results obtained for the method are an effect of unduly restricting weight-optimized network to less-efficient evolutionary algorithms?

**Summary Of The Paper:**

The paper augments standard neural networks with recurrent, parametrizable "activation functions". They show that randomly initialized networks with frozen weights can be trained with some degree of success by simply optimizing the parameters of these "activation functions". The results are competitive with normal, weight-optimized neural networks with similar number of parameters, and also in some cases with neural networks with an equivalent number of weights (and thus many more parameters), when trained with evolutionary algorithms.

**Summary Of The Review:**

To my understanding, the  actual method is quite different from what is advertised. The  paper says that optimizing activation functions alone produces competitive results. The actual finding is that adding trainable *recurrence* to frozen-weight networks shows non-trivial performance on a specific type of tasks (dynamic control).  This might be of some interest, but even then, the experiments may not fully reflect the actual performance in comparison to trained-weight networks with modern RL algorithms.

---

### Official Review · Reviewer_ah36 · 2022-10-25

**Confidence:** 4
**Correctness:** 4
**Technical Novelty And Significance:** 2
**Empirical Novelty And Significance:** 2
**Recommendation:** 3

**Clarity, Quality, Novelty And Reproducibility:**

The text is well-written; it contains sufficient detail for the experiments to be reproduced. The quality of the experiments can be strengthened by using standard, more challenging baselines.

**Strength And Weaknesses:**

The goal of designing a new neural network architecture is worthy and timely, as it extends a growing body of work on network architectures including the Lottery Ticket Hypothesis and the Weight-Agnostic Networks, known to NeurIPS/ICML communities and thoroughly referenced in the text. The model has been tested on a standard set of reinforcement-learning benchmarks, allowing us to compare the results to those from other models. The new approach is complementary to the existing models and can be easily combined with them.

If I’m not missing anything, the message of the paper is that the proposed architecture, while being different from the existing ones, shows similar performance. While the result is interesting in its own right, more analysis is typically expected in ICLR submissions to assess the scope and the usefulness of the proposed model. Specifically, what are the possible use cases where applying such a model would be advantageous? If the main benefit of this model is in the reduced number of parameters, then a comparison with up-to-date compression approaches is needed. Should the main benefit of the model be in the improved performance, a comparison with the top-scoring models would be necessary for the considered RL tasks. For now, the best-reported performance of the models does not reach the current standards*, and in two out of three tests the best performance was not achieved with the proposed model.

* huggingface.co/spaces/huggingface-projects/Deep-Reinforcement-Learning-Leaderboard

I understand that the authors aimed to compare the models in a simple setting and did not have a goal of exceeding the current best results in selected RL tasks, however, the possible advantages of the proposed model appear most important for large high-performance networks and not for small networks that don’t need to be compressed and whose performance may be improved by the conventional means. It is not obvious if the reported results generalize to more complex high-performance networks.

Minor comments:

-As the proposed model uses memory and matrix multiplication to define the activation function, it is important to show that both components are needed for the reported performance.

-It would be also great to have a description of how the proposed approach is different from (a subclass of) RNNs which also have memory and use matrix products before the nonlinearity.

-For the Weight-Agnostic Networks, the number of parameters can be estimated as the number of coordinates of nonzero connections, i.e. the size of the netlist representation of the graph.

**Summary Of The Paper:**

The authors propose an approach to the training of neural networks where, instead of adjusting weights between neurons, they adjust nonlinearities on the output of each neuron. The nonlinearities are applied via a product between a learnable matrix and a vector consisting of the neuronal input, an updateable memory term, and a bias term – all followed by a fixed nonlinear function. The authors test their model on three standard RL tasks and find that the model’s performance is similar to that of the conventional neural networks with a matching number of layers/parameters.

**Summary Of The Review:**

Although the paper pursues an important goal of introducing a new neural architecture for deep learning, additional experiments are needed to show what are the advantages of using this architecture. Until such experiments are performed, the results deem preliminary for ICLR, yet may be sufficient to be reported elsewhere, e.g. as a preprint.

---

### Official Review · Reviewer_3uev · 2022-10-25

**Confidence:** 5
**Correctness:** 2
**Technical Novelty And Significance:** 2
**Empirical Novelty And Significance:** 1
**Recommendation:** 3

**Clarity, Quality, Novelty And Reproducibility:**

The paper is for the most part clearly written and describes the experiments in a good level of detail. To the best of my knowledge the work is original.

**Strength And Weaknesses:**

Strengths: The paper is generally well written, and describes it main idea clearly along with the corresponding experiments.

Weaknesses: I have concerns with the motivation and architectural designs for how this paper implements its main idea, as well as with the overall impact of the results.

- Motivation: The paper introduces the idea of diversity in neuroscience (specifically diverse cell types), and uses this to motivate the need for diverse activation functions in artificial networks. I find this connection pretty weak and uninspiring. I want to know _why_ we see cell type diversity in biology, specifically what is the computational benefit from having different cell types (this has been studied and discussed for many different brain regions in the neuroscience literature). Unsurprisingly, the _reasons_ for cell type diversity are as diverse as the cell types themselves. Furthermore, the mechanistic implications of cell type diversity extend beyond just activation functions (this too has been discussed heavily in the neuroscience literature). Overall, I found the motivations this paper takes from neuroscience to be weak and too vague to be useful.

- Architectural designs: The proposed architecture, to me, looks very different from an "activation function". When I hear that term, I think of a point-wise (1D) nonlinearity in a neural network. Here, the paper defines an activation function as a small recurrent unit! Equation (1) in the paper looks just like a small vanilla RNN. That's fine (exploring what happens when you augment neurons with recurrent state is interesting), but this implementation is too different (in my mind) from the focus on "activation functions" in the title and abstract.

- Memory: One big difference between the introduced architecture and the feedforward baselines is that the new architecture has memory. This is discussed briefly, but to me an important control is what happens if you remove the memory mechanism from the network? That is, are the observed improvements just due to having memory (state)? I suspect so, and if so, the title/abstract/introduction should be rewritten to emphasize this.

Suggestions for improvement:
- Better clarify the network architecture. How is it different from an RNN? What is the role of memory?
- Are the performance differences across networks due to a failure to optimize, or the fact that the networks have different computational capacities? I.e. is it possible to find a setting of the parameters that can solve the task, but it is difficult to do so with the given optimization method?
- More thorough hyperparameter comparisons are necessary to understand the limitations of the method.
- Can we develop some deeper understanding of what these networks learn, and how that differs from both standard feedforward networks and standard recurrent networks? This is crucial to know how interesting these networks are.
- Better clarify the connection between the motivation (diversity in biological networks) and these architectural choices. At the moment, reads like “biology is complicated, so let’s try complicated networks”—which is fairly shallow.

**Summary Of The Paper:**

This paper introduces a type of neural network architecture that augments individual units in the network with an additional state variable and reparameterizes them using a 3x3 weight matrix. Each individual unit receives input that is the output of a linear transformation (as in standard networks), but instead of passing that output through a pointwise nonlinear activation function the inputs are transformed through a small state update equation. The network compares the performance of these networks against feedforward networks on three RL tasks.

**Summary Of The Review:**

Overall, I had issues with the motivation and implementation of the main idea. I think changing the framing to focus on the interesting bit of the architecture (adding recurrent memory to individual units), and further clarifying the neuroscientific connections (are there examples in neuroscience of mechanisms within neurons that can implement recurrent state, and how those might lead to computational benefits?) are needed.

---

### Official Review · Reviewer_xrLZ · 2022-10-25

**Confidence:** 3
**Correctness:** 2
**Technical Novelty And Significance:** 2
**Empirical Novelty And Significance:** 2
**Recommendation:** 3

**Clarity, Quality, Novelty And Reproducibility:**

The methods description is unclear. While learning activation functions in not novel, the specific model-form here appears novel to me.

**Strength And Weaknesses:**

Strengths
- Learning via activation functions is an interesting and important direction of research, as is the connection to biological neuron response functions.
- Paper is generally well written.

Weaknesses
- I disagree that the model presented here is best described as “learning activation functions”, as a parametric activation function is not being learned. Instead the model is in the space of models that could be realized by a standard RNN.
- No biological justification is given for the model’s form. I think the biological motivations and relevance of this paper are overstated. How neural firing types or classes are related to what is learned by the network is unclear and not addressed.
- No justification is given for why RL is an appropriate task that this model is either specifically suited for, or to test properties of the model.
- Related work section is not as relevant as Sipper, Agostinelli and Bingham papers referenced in the introduction.
- Why are weights sampled a gaussian with 0.5 std?
- Synergies with standard weight optimization should be demonstrated, currently no performance benefit is presented. Results are underwhelming.
- Methods & implementation details unclear and no link to code.



**Summary Of The Paper:**

Inspired by classes of biological neurons, the authors explore if RNNs can be trained to perform simple RL tasks via neuron activation function optimization.


**Summary Of The Review:**

While an interesting research direction, I do not think that an activation function is being learned here and the biological connections are undeveloped. Results are limited and underwhelming.

---

### Decision · Program_Chairs · 2023-01-20

**Decision:**

Reject

**Justification For Why Not Higher Score:**

The paper does not really deliver on its promise, and there was no attempt to rebut the reviewers. With an average score of 3.5 there is clearly no way this paper should be accepted.

**Justification For Why Not Lower Score:**

N/A

**Metareview: Summary, Strengths And Weaknesses:**

This paper explores the possibility of training neural networks using only modification of the activation functions of the neurons. The authors demonstrate that this can work with some basic control tasks.

The strengths of this paper are that it is reasonably clear and the core idea is interesting.

However, the reviewers noted that this paper is arguably misrepresenting the model by portraying this as "activation function only" optimization, given that each neuron is now treated like an recurrent network unto itself, meaning that the claim that it is "only" activation functions being tuned depends on an odd interpretation of the network structure. As well, there were concerns about the appropriateness of some of the empirical tests.

The authors did not respond to the reviewers' comments, and the average score was 3.5, making this a clear reject.

**Summary Of Ac-Reviewer Meeting:**

N/A